# Tired Actor: Fatigue-Informed Character Control

## Abstract

Replicating human behavior with physics simulation has been a long-expected goal in character animation. Existing efforts have achieved impressive performance in imitating a wide span of general motions. However, most existing efforts could still suffer from **unnatural movements** due to the lack of biomechanical and physiological priors. Given this, we project our sights to advances in behavioral energetics, which demonstrate how energy use shapes human movements. In contrast, current character controllers typically assume the character is equipped with infinite energy over time. Inspired by these, we propose to adopt **fatigue** as a proxy of the finite energy limit, inject it into general character animation, and thoroughly investigate how fatigue introduces new characteristics to physics-based character control. Leveraging the Three-Compartment Controller (3CC) model, we managed to obtain a policy for general motion imitation under different fatigue statuses. Furthermore, extensive analyses are conducted to demonstrate how fatigue could influence the naturalness, scalability, and robustness of character animation. *Our code would be made public.*

## 1 Introduction

Physically controlling human-like characters to produce natural motions has been a long-pursued goal for the vision and graphics community, given its great potential in character animation (Peng et al., 2022; Zhu et al., 2023), virtual reality (Winkler et al., 2022; Luo et al., 2024b), and robotics (He et al., 2025b). Tremendous progress has been achieved in reproducing human motions with simulated characters (Luo et al., 2023; Truong et al., 2024). Moreover, deployments on humanoid robots are also made possible with impressive agility (He et al., 2025a) and expressiveness (Cheng et al., 2024).

Despite the impressive advances, existing efforts still produce *unnatural movements under certain circumstances*, as shown in Fig. 1, since most are purely data-driven with limited biomechanical and physiological priors. Given this, we project our sight to an emerging research field termed behavioral energetics (McAllister et al., 2025), which investigates how energy use shapes human movements. Researches demonstrate how energy consumption influences the gaits and speed during locomotion (Ralston, 1958; Brooks et al., 2005), gait transition (Diedrich & Warren Jr, 1995; Mercier et al., 1994), cycling cadence (Brisswalter et al., 2000; Riveros-Matthey et al., 2025), and navigation in dynamic contexts (Brown et al., 2021). These efforts reveal that natural human movement patterns are tightly bound with *energy optima*. Therefore, introducing energy sensing or its proxies to character control becomes a promising way to improve the naturalness. In contrast, most efforts assume the character is equipped with *infinite energy*. The assumption's downside has not been noticed much, which is a side effect of the typical task formulation, where more attention is paid to short motion sequences. The tracked motion sequences end before the cumulative tiredness could make an impact. However, with the controllers becoming more and more satisfying, high hopes could be placed on the character control polices for lifelike movements with advanced applications like motion generation and stylization. In this way, the human-likeness of characters could be a critical goal, and the infinitely energetic character is lower than expected.

Inspired by these, we propose to leverage fatigue as a proxy of energy-awareness and incorporate it with state-of-the-art character controllers, expecting to maintain the motion coverage while improving the naturalness of the character with behavioral energetics clues as in Fig. 1. A Three-

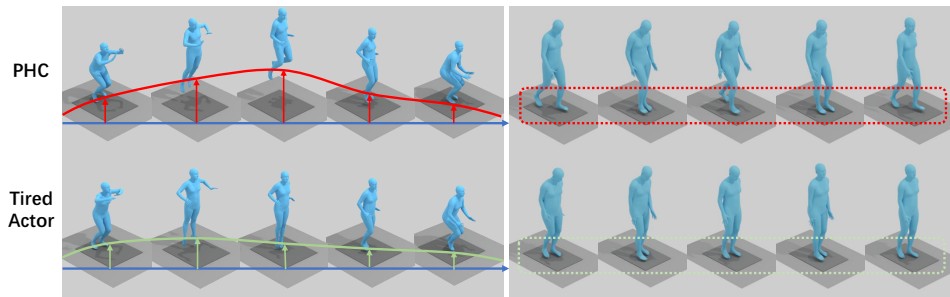

Figure 1: Current controllers could still be unnatural, replicating treadmill paces on the ground. Inspired by behavioral energetics, we introduce Tired Actor, a fatigue-informed character controller. As shown, our Tired Actor learns to cut corners by reducing action amplitudes. It also produces a reasonable and natural single-leg swinging motion instead of foot sliding on the ground, which is an interesting side effect.

Compartment Controller (3CC) model (Xia & Law, 2008) adapted for torque-based motion control (Cheema et al., 2023) is adopted to represent the human fatigue mechanism, where the effect of fatigue is simulated via modifying the maximal applicable torque. In detail, we first collect the maximal applicable torques by rolling out existing polices. Then, a fatigue-informed policy, Tired Actor, is trained concerning the collected maximal torques. Moreover, we conduct thorough experiments on how fatigue reshapes the behaviors of the controlled character. Specifically, we focus on three aspects. First, different fatigue states could result in different motion states not in the original data distribution, and encourage the policy to produce novel behaviors. Relatedly, we are also curious about how the fatigue-extended data distribution might help the policy generalize to unseen behaviors. Finally, inspired by the relationship between fatigue and compensation in human motion, we are also interested in whether fatigue-informed policy is equipped with proper compensation.

In summary, our main contribution is twofold: First, inspired by behavioral energetics insights, we incorporate fatigue with a character policy termed Tired Actor, improving its naturalness while maintaining the motion coverage. Second, we conduct thorough experiments to investigate how the fatigue mechanism could influence the diversity, generalization, and compensation.

## 2 RELATED WORKS

**Physics-Based Character Control.** Physics-based character control has been an active research field. Early advances focused on tracking controllers for specific and limited motions (Geijtenbeek et al., 2013; Liu et al., 2010). With the development of deep RL algorithms, success was achieved for imitating motion sequences with manually designed rewards (Liu & Hodgins, 2017; Peng et al., 2018). To reduce the reliance on manual engineering, Merel et al. (2017); Peng et al. (2021) introduced GAIL (Ho & Ermon, 2016) for motion imitation. However, the generalization was still limited. With residual force control (Yuan & Kitani, 2020), Luo et al. (2021) managed to replicate over 97% motion sequences in AMASS (Mahmood et al., 2019), achieving noticeable generalization ability. Luo et al. (2023) eliminated the supernatural residual force while maintaining the generalization ability. Truong et al. (2024) leveraged diffusion policy (Chi et al., 0) for better recovery ability. With the well-developed motion-tracking controllers, controllers for more advanced tasks have become a new focus. Peng et al. (2022) proposed to learn skill embeddings for character controllers. Xu et al. (2023) learned decoupled skill representations for different body parts. Towards better modeling of human skills, VAEs (Yao et al., 2022; Luo et al., 2024c; Tessler et al., 2024), VQ-VAEs (Zhu et al., 2023; Yao et al., 2024), and diffusion models (Truong et al., 2024) are adopted to regulate the skill space. Different high-level control signals, including text prompts (Juravsky et al., 2022; Kumar et al., 2023; Ren et al., 2023; Juravsky et al., 2024; Tevet et al., 2025), VR controller signals (Winkler et al., 2022; Luo et al., 2024b), and high-level action semantics (Tessler et al., 2023; Dou et al., 2023) are also incorporated to character control. Emerging efforts started to pay more attention to the interaction ability of the controlled actor. Luo et al. (2022) leveraged a simulated character with scene-awareness for accurate 3D pose reconstruction. A series of works controlled characters to produce feasible movements in different scenes (Hassan et al., 2023; Lee

et al., 2023; Xiao et al., 2024), even producing highly agile parkour movements (Xu et al., 2025a). Moreover, controlling characters to interact with objects also gained attention. Early efforts focused on graspable objects (Braun et al., 2024; Luo et al., 2024a), while recent advances grew the capability in whole-body manipulation (Gao et al., 2024; Xu et al., 2024; Yu et al., 2025; Xu et al., 2025b) and sport skills (Wang et al., 2024; Yu et al., 2025). Despite the impressive advances, most existing efforts were built on a purely data-driven basis, with limited exploration of biomechanical and physiological priors. Unnatural movements could be produced under certain circumstances, as shown in Fig. 1. Given this, we resort to behavioral energetics for the physiological prior.

**Behavioral Energetics.** Traditional views of biomechanics take the energetic cost as the result of human mechanics (Donelan et al., 2004; Gottschall & Kram, 2003; Arellano & Kram, 2011). Instead, behavioral energetics suggests that energetic cost plays an important role in shaping human movements. Research has shown that humans tend to move with minimal energy use (Atzler & Herbst, 1927; Elftman, 1966; Alexander, 1996). Calories per unit distance are optimized concerning different walking speeds by humans (Molen et al., 1972; Ralston, 1958). Gait parameters like step frequency and width are also energetically optimal (Bertram & Ruina, 2001; Holt et al., 1991). The energetic optimum also appears in pace selection for sustainable running (Rathkey & Wall-Scheffler, 2017; Steudel-Numbers & Wall-Scheffler, 2009; Willcockson & Wall-Scheffler, 2012). Some research also suggests the optima could be developed and refined via training (Cher et al., 2015). Humans switch between walking and running at a speed at which it becomes energetically more expensive to walk than to run under treadmill settings (Diedrich & Warren Jr, 1995; Hreljac, 1993). In more natural settings, humans select among walking, running, and resting with dynamic speed adaptation to minimize the energy usage concerning the distance requirements and time limits (Long III & Srinivasan, 2013). New evidences show energy optimality could even explain how humans select path trajectories for turning and obstacle avoidance (Brown et al., 2021; Daniels & Burn, 2023). Given these advances, we identify behavioral energetics as a promising inspiration for natural character control and introduce human fatigue as a proxy for energy sensing.

**Human Fatigue Modeling.** Giat et al. (1993; 1996) developed fatigue models for the quadriceps from fatigue-related metabolic parameters during functional electrical stimulation and recovery, while Ding et al. (Ding et al., 2000) introduced a four-parameter model to predict musculoskeletal fatigue. Despite their realism, they were heavily based on detailed muscle activation patterns, limiting their application. To this end, a motor unit-based fatigue was proposed (Liu et al., 2002), where the fatigue is estimated via three muscle states: activated, fatigued, and resting. Based on this, the Three-Compartment Controller (3CC) model (Xia & Law, 2008) was introduced for dynamic load conditions. Cheema et al. (2023) further extended it for torque-based scenarios. There were also efforts in developing a data-driven model for human fatigue (Kider et al., 2011), where a multi-modal capturing technique was developed. Even less effort was paid to fatigue-aware character control. Early efforts were made on human body parts, either for lower-body motion control (Komura et al., 2000) or single-arm motion control (Cheema et al., 2020). Feng et al. (2023) developed a fatigue-aware motion controller for muscle-actuated characters. Concurrently, Cheema et al. (2023) enhanced AMP with fatigue. However, both were limited in the scale of applicable motion sequences. Also, the influence that the fatigue mechanism could impose has not been structurally investigated.

## 3 METHOD

### 3.1 FATIGUE-INFORMED CHARACTER CONTROL PRELIMINARIES

We formulate our character controller $\pi(a^t | s_{task}^t, s_t^{self})$ w.r.t. the Markov Decision Process $\mathcal{M} = \langle \mathcal{S}, \mathcal{T}, \mathcal{A}, \mathcal{R}, \gamma \rangle$, composed of states $\mathcal{S}$ governed by the transition dynamics $\mathcal{T}$, available action space $\mathcal{A}$, reward function $\mathcal{R}$, and discount factor $\gamma$. At each timestep $t$, given the current state $s^t$ and target state $s^{task}$, the control policy $\pi$ produces action $a^t \in \mathcal{A}$. With transition dynamics $\mathcal{T}$, time proceeds, producing the state of the next timestep $s^{t+1}$. Reward $r$ is then computed following the reward function $r^t = \mathcal{R}(s^{task}, s^{t+1}, s^t, a^t)$. The training goal is to maximize the reward expectation $\mathbf{E}(\sum_{t=1}^{T} \gamma^{t-1} r^t)$. Tired Actor, our trained controller, is designed to combine PHC (Luo et al., 2023) and Cheema et al. (2023) to maintain the motion coverage while injecting the fatigue mechanism.

**States $\mathcal{S}$.** A 24-joint SMPL (Loper et al., 2015) humanoid is adopted with full-zero body shape parameters. At each time step, the state $s^t$ could be decoupled into two separate components: the self-

state $s^t_{self}$ and the task state $s^t_{task}$. The self-state $s^t_{self}$ contains character proprioception information $s^t_{self} = \{h^t_{root}, x^t, \dot{x}^t, \theta^t, \dot{\theta}^t, M^t_F\} \in R^{427}$. $h^t_{root} \in R$ represents the root height. $x \in R^{69}, \dot{x} \in R^{72}$ are the positions and linear velocities of different body keypoints, canonicalized in the egocentric coordinate. Therefore, the position of the root joint is eliminated from $x^t$. $\theta \in R^{144}, \dot{\theta} \in R^{72}$ are the orientations and angular velocities of different body links, also canonicalized in the egocentric coordinate. Finally, $M^t_F \in [0, 100]^{69}$ indicates the fatigue level at timestep $t$, which will be covered later. The task state is defined as:

$$s^t_{task} = \{\hat{x}^{t+1} - x^t, \hat{\dot{x}}^{t+1} - \dot{x}^t, \hat{\theta}^{t+1} - \theta^t, \hat{\dot{\theta}}^{t+1} - \dot{\theta}^t, \hat{\theta}^{t+1}, \hat{x}^{t+1}\} \in R^{576}, \quad (1)$$

where $-$ indicates the rotation difference for $\theta, \dot{\theta}$. Quantities with $\hat{}$ are the corresponding next-step states in the reference motion. All the quantities are in the egocentric coordinate.

**Action $\mathcal{A}$.** All joints but the root (pelvis) are actuated with proportional derivative (PD) controllers. The action $a^t$ is composed of two components as $a^t = \{\mu^t, \beta^t\} \in R^{70}$, where $\mu^t$ is the PD target, and $\beta^t$ is a stiffness and damping multiplier following (Cheema et al., 2023), which is expected to tune the tension of the whole body. In practice, we also find that this decoupled action design makes it easier for the policy to adapt to different fatigue levels. Without introducing the fatigue mechanism, the raw applied torque $\tau^t_{raw}$ could be computed as

$$\tau^t_{raw} = \beta^t(k_p \circ (\mu^t - q^t) - k_d \circ \dot{q}^t), \quad (2)$$

where $k_p, k_d$ are the stiffness and damping factors.

**Transition Dynamics $\mathcal{T}$.** We adopt IsaacGym (Makoviychuk et al., 2021) for rigid body dynamics simulation. The transition dynamics are handled by the 3CC model (Liu et al., 2002; Xia & Law, 2008; Cheema et al., 2023), which assumes motor units are either active ($M_A$), fatigued ($M_F$), or resting ($M_R$), represented in the percentage of maximum voluntary torques. Initially, all motor units are resting ($M_R = 100\%$). Resting units become activated once a target load ($TL$) appears, resulting in increased $M_A$. Activated units could then be fatigued over time, with descending $M_A$ and increasing $M_F$. Meanwhile, the fatigued units could also regain their power over time, resulting in $M_F$ decreasing and $M_R$ increasing. The transition dynamics of fatigue could be described as

$$\frac{\partial M_A}{\partial t} = C(TL) - F \cdot M_A, \frac{\partial M_F}{\partial t} = F \cdot M_A - R_r \cdot M_F, \frac{\partial M_R}{\partial t} = -C(TL) + R_r \cdot M_F, \quad (3)$$

$$R_r = \begin{cases} R \cdot r, & \text{if } M_A \geq TL, \\ R, & \text{else,} \end{cases} \quad (4)$$

$$C(TL) = \begin{cases} L_R \cdot (TL - M_A), & \text{if } M_A \geq TL, \\ L_D \cdot (TL - M_A), & \text{if } TL - M_R < M_A < TL, \\ L_D \cdot M_R, & \text{else.} \end{cases} \quad (5)$$

Here, $F, R$ control the fatigue and recovery rates. $1 - \frac{R}{F}$ also indicates the upper bound of $M_F$. $r$, as an additional rest recovery multiplier for intermittent tasks (Looft et al., 2018). $r > 1$ indicates that the recovery is faster when not all active units are necessary to achieve the target load (TL). $C(TL)$ is a bounded proportional controller, producing the required TL while dynamically changing the proportion of $M_A$ and $M_F$. $C(TL)$ is characterized by muscle development factor $L_D$ and relaxation factor $L_R$, preventing $M_A$ and $M_F$ from instantaneous changes, thus simulating the muscle activation dynamics (Thelen, 2003; Winters, 1995). With this, the applied torque in Sec. 3.1 is further modified according to the fatigue state as

$$TL = \frac{\tau^t_{raw}}{\tau_{max}}, RC = 1 - M_F, \tau^t = clip(\tau^t_{raw}, -RC \cdot \tau_{max}, RC \cdot \tau_{max}). \quad (6)$$

Here, $\tau_{max}$ represents the maximal exertable torque, and $\tau^t$ is the final applied torque.

**Rewards.** We define the reward function as $r_t = 0.5r^t_{task} + 0.5r^t_{amp} + r^p_t + r^f_t$. In detail, the task reward is defined as $r^t_{task} = \omega_1 e^{-100\|\hat{x}^t - x^t\|} + \omega_2 e^{-0.1\|\hat{\dot{x}}^t - \dot{x}^t\|} + \omega_3 e^{-10\|\hat{\theta}^t - \theta^t\|} + \omega_4 e^{-0.1\|\hat{\dot{\theta}}^t - \dot{\theta}^t\|}$, where the translation and linear velocities of all joints, and the orientation and angular velocities of all links are considered. For reward $r^t_{amp}$, we follow AMP (Peng et al., 2021) for the observations, loss formulation, and gradient penalty design. For power reward $r^t_p$, we follow (Luo et al., 2023) as $r_p = -0.0005|\dot{\theta}^t \tau^t|^2$, encouraging low-energy movements which tend to look more natural (Yu et al., 2018). We further introduce a fatigue reward $r^t_f = 0.01M^t_F$ for explicit regulation on cumulative fatigue minimization.

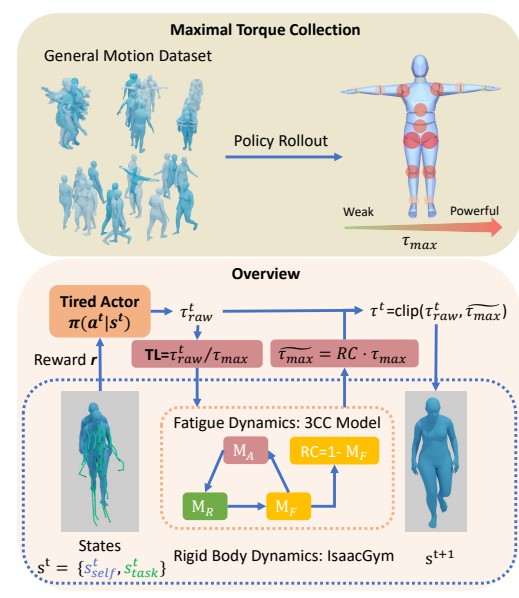

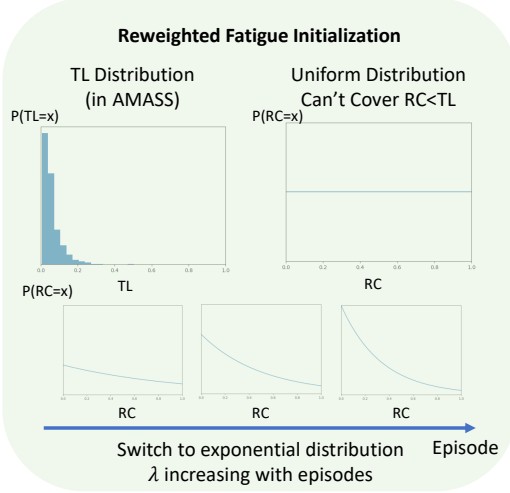

Figure 3: Reweighted fatigue initialization. Due to the imbalanced fatigue distribution in general human motion, instead of uniformly sampling fatigue levels in training, we adopt an exponential distribution to encourage the policy to learn fatigue-related motion patterns.

Figure 2: Tired Actor pipeline. We first collect the maximal torque limits from AMASS. Then, we train the Tired Actor with fatigue-awareness.

### 3.2 TIRED ACTOR: FATIGUE-INFORMED CHARACTER CONTROLLER

Next, we introduce our fatigue-informed controller: Tired Actor. It is simply designed as $\pi(a^t|s^t) = \mathcal{N}(\mu(s_t), \sigma(s_t))$, where $\mu(\cdot), \sigma(\cdot)$, and the discriminator for $r_{amp}$ are all MLPs. We first collect maximal torque limits by rolling out existing motion controllers. Then, the Tired Actor is trained with reweighted fatigue initialization, hard negative mining, and fall-recovery finetuning. The general pipeline is shown in Fig. 2.

**Maximal Torque Collection.** A key component for the fatigue-informed controller is the maximal torque $\tau_{max}$, which defines the maximal applicable torque. Previous effort (Cheema et al., 2023) pre-trained per-sequence motion controllers to obtain per-action torque limits. This might be preferable for single-action control. Instead, aiming at making Tired Actor compatible with general motion sequences, we directly roll out PHC on AMASS (Mahmood et al., 2019), and record the maximal torques for each joint according to Eq. 2 with $\beta^t = 1$. Thus, the obtained torque limits could function more like a representation of general human motor capacity than an action-intensity representation.

**Reweighted Fatigue Initialization.** With the collected torque limits, we could now train our fatigue-informed controller with PPO. The criterion network shares the same structure as the policy. At the start of each episode, the initial fatigue states $M_F, M_A, M_R$, and fatigue parameters $F, R, r$ are randomly sampled. However, this introduces new problems. In Cheema et al. (2023), $M_F, M_A, M_R$ was uniformly sampled. However, our general-purpose torque limits could be much higher than single-action limits. For most actions, $TL$ is less than $10\%$, meaning that even $M_F = 90\%$ could make little difference in motion patterns. Thus, the original uniform distribution could not cover the $RC < TL$ scenario well, which is important for learning fatigued movement patterns. Therefore, we introduce reweighted fatigue initialization as shown in Fig. 3. Instead of uniform distribution for $RC(= 1 - M_F)$, we first sample $RC$ with exponential distribution $Exp(\lambda_{ep})$ whose parameter $\lambda_{ep}$ increases with the number of episodes, denoted as

$$RC \sim \exp(\lambda_{ep}), M_F = 1 - \text{clip}(\text{RC}, 0, 1). \tag{7}$$

In this way, we make more explorations in the high-fatigue region, encouraging the policy to focus on fatigue-related motion patterns. Meanwhile, the varying $\lambda_{ep}$ ensures a smooth learning procedure, preventing the model from failing in the early stage due to difficult, high-fatigue scenarios.

**Hard Negative Mining.** Hard negative mining is a proven-successful technique in previous efforts Luo et al. (2021; 2023). During training, we regularly evaluate our trained controller on the whole dataset and identify the hard sequences as the sequences in which our controller fails. The

| Method | AMASS-Train | | | | | AMASS-Test | | | | |
|---|---|---|---|---|---|---|---|---|---|---|
| | Success Rate (%) | $mPJPE-G$ | $mPJPE-L$ | Acc. Error | Vel. Error | Success Rate | $mPJPE-G$ | $mPJPE-L$ | Acc. Error | Vel. Error |
| PHC | 98.9 | 37.5 | 26.9 | **3.3** | **4.9** | 96.4 | 47.4 | 30.9 | 6.8 | 9.1 |
| Baseline | 98.5 | **34.7** | 27.0 | **3.3** | **4.9** | 96.4 | 51.8 | 32.3 | 6.0 | 8.4 |
| Tired Actor | **99.1** | 37.4 | **25.4** | 3.7 | 5.1 | **97.9** | 47.6 | **30.1** | **5.7** | **8.2** |

Table 1: Results of motion imitation on AMASS.

| Parts | Baseline | | Tired Actor | |
|---|---|---|---|---|
| | mPJPE-G | mPJPE-L | mPJPE-G | mPJPE-L |
| Pelvis | 30.2 | - | **28.6** | - |
| Hips | 32.3 | 9.1 | **30.2** | **8.8** |
| Knees | **42.5** | **33.8** | 45.6 | 38.8 |
| Ankles | 60.2 | 53.2 | **54.4** | **51.9** |
| Shoulders | **30.1** | 19.5 | 31.4 | **18.5** |
| Elbows | **31.4** | 28.5 | 33.8 | **26.0** |
| Wrists | **35.1** | 36.7 | 37.7 | **31.9** |

Table 2: Part-level mPJPE comparison on AMASS-Train with initial $M_F = M_A = 0, M_R = 100$.

sampling weights of these hard sequences are increased. By failure, we follow the definition in (Luo et al., 2021; 2023) as the state with mPJPE greater than 0.5 meters.

**Fall-recovery Finetuning.** In PHC, the fall-recovery ability is achieved by training an extra primitive for recovery and ensembling it with other primitives. Instead, we directly fine-tune our controller with fall-recovery scenarios after achieving general motion imitation abilities. We follow a simplified design of that in (Luo et al., 2023). At the beginning of randomly decided episodes, we simulate without applying any torques for a certain time step, making the character fall. Then, we start training from the fallen states. We did not adopt the fallen-state reward in (Luo et al., 2023) due to our limited controller capacity. Even though we observe an increasing ability in hard sequence imitation and fall recovery with the finetuning procedure.

## 4 EXPERIMENTS

We train Tired Actor on the same AMASS (Mahmood et al., 2019) training split as PHC (Luo et al., 2023). More implementation details are available in the appendix.

### 4.1 FATIGUE-INFORMED MOTION IMITATION

We first evaluate the performance of Tired Actor on motion imitation. Following PHC (Luo et al., 2023), we report the Success Rate. When mPJPE$> 0.5m$, we deem the imitation as failed and terminate the imitation. The global mPJPE-G and root-relative mPJPE-L are both reported in millimeters. Also, the acceleration error and velocity errors are reported. Besides PHC, we also train a baseline by eliminating the fatigue modeling of the Tired Actor. The fatigue parameters of Tired Actor are $F = 2, R = 0.05, r = 1$. The initial fatigue state is set as $M_F = M_A = 0, M_R = 100$. Quantitative results are shown in Tab. 1. On the AMASS train split, Tired Actor achieves the highest success rate. The advantage of Tired Actor is also seen on mPJPE-L. However, Tired Actor is noticeably worse than baseline on mPJPE-G, acceleration, and velocity imitation. These indicate fatigue might encourage the sacrifice of the global pose following ability in exchange for the local pose imitation ability. Also, the velocity and acceleration are giving way to the optimization of mPJPE-L. For the unseen AMASS test split, Tired Actor surprisingly provides better generalization ability, surpassing the baseline on all metrics. A more detailed analysis of per-part mPJPE is provided in Table 2. The tradeoff is not only involving global and local tracking, but also entangled with the lower-body and upper-body. Noticeably, the Tired Actor is superior in following the root trajectory. The knee joint, as the key source of lower-body agility, trades its tracking precision for the precision of hips and ankles. For the upper body, Tired Actor typically prefers local pose resemblance. We further demonstrate some qualitative samples in Fig. 4. The Tired Actor learns to cut corners when imitating the reference motion by reducing action amplitudes and trailing behind. With the initial $M_F$ increasing, the model tends to put less effort and exhibits smaller action amplitudes. We also compute the step length of the Tired Actor and baseline on 500 walking sequences, where the Tired Actor presents a noticeably lower step length of 0.385 m compared to the baseline with 0.407 m. Interestingly, the cutting-corner behavior could improve the motion naturalness under certain scenarios, as shown in Fig. 1. Instead of stubbornly imitating the treadmill walking motion on static ground like PHC, Tired Actor swings a single leg to balance between resemblance and naturalness.

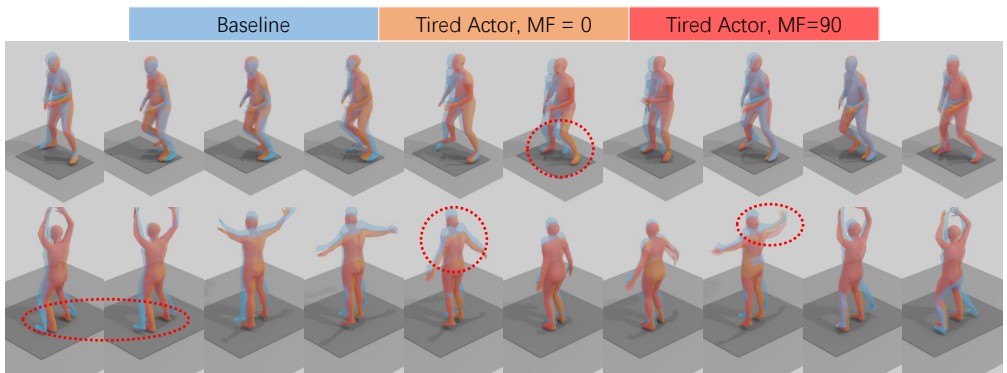

Figure 4: Motion imitation with different initial $M_F$s.

| Method | AMASS-Train | | | | | AMASS-Test | | | | |
|---|---|---|---|---|---|---|---|---|---|---|
| | Success Rate | mPJPE-G | mPJPE-L | Acc. Error | Vel. Error | Success Rate | mPJPE-G | mPJPE-L | Acc. Error | Vel. Error |
| Tired Actor ($M_F$=0) | **99.1** | **37.4** | **25.4** | 3.7 | 5.1 | **97.9** | 47.6 | 30.1 | **5.7** | **8.2** |
| Tired Actor ($M_F$=50) | 98.4 | 37.4 | 26.5 | **3.4** | **5.0** | 97.1 | **46.9** | **29.7** | 5.8 | 8.3 |
| Tired Actor ($M_F$=80) | 98.0 | 38.0 | 27.2 | 3.5 | **5.0** | 95.7 | 47.7 | 30.1 | 5.8 | 8.3 |
| Tired Actor ($M_F$=90) | 96.9 | 39.1 | 28.0 | 3.6 | 5.2 | 94.3 | 49.3 | 31.4 | 6.0 | 8.6 |
| Tired Actor ($M_F$=95) | 91.1 | 44.0 | 31.1 | 4.7 | 6.1 | 89.3 | 52.5 | 33.6 | 6.7 | 9.3 |

Table 3: Motion imitation with different initial $M_F$s.

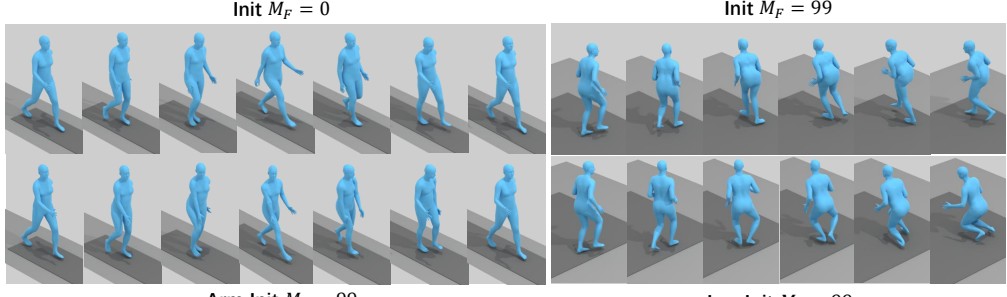

Figure 5: Fatigue-induced cross-part coordination.

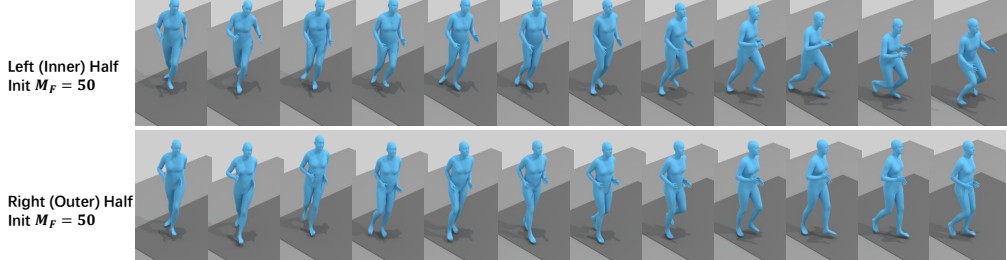

Figure 6: Fatigue-induced asymmetry. Fatigue on inner-/outer-half body results in diverse behaviors.

## 4.2 FATIGUE-INFORMED BEHAVIORS

**Different Initial $M_F$.** Besides the qualitative demonstrations in Fig. 4, quantitative results are reported in Tab. 3 with shared fatigue parameter $F = 2, R = 0.05, r = 1$. $M_F$ introduces a marginal performance drop until reaching about $90\%$, consistent with the fact that most movements only need less than $20\%$ maximum torque. Also, not all the metrics reach the best level with the initial $M_F = 0$. It might be easier to learn some motions with "weaker" characters.

**Different Body Parts.** Fig. 5 illustrates how fatigue of different parts could influence character motion. When initializing the arms with $M_F = 99\%$ and others with $M_F = 0$, the arms only experience a short period of stiffness with a quick recovery. This indicates that fatigue of distal segments could be easily handled. We also initialize the legs with $M_F = 99\%$ and others with zero $M_F = 0$, compared to a character with whole-body initial $M_F = 99\%$. The former falls due to a strength disorder in different joints. Instead, the latter balances by coordinating the weak but consistent joints, showing the critical role of consistency for fatigue compensation.

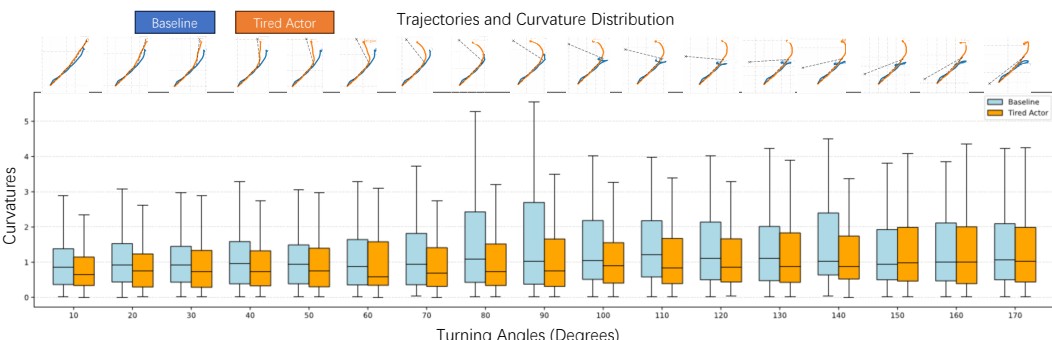

Figure 7: Fatigue-induced spatial decision making. Trajectories and curvature distributions when faced with different turning angles are visualized. Tired Actor typically avoid sharp turns with lower curvatures, traded-off with the target reaching speed for rather sharp turns.

| | AMASS-Train | | | AMASS-Test | | | CIRCLE | | |
| Method | Success Rate | mPJPE-G | mPJPE-L | Success Rate | mPJPE-G | mPJPE-L | Success Rate | mPJPE-G | mPJPE-L |
| --- | --- | --- | --- | --- | --- | --- | --- | --- | --- |
| Baseline (100 seq) | 85.8 | **43.1** | **32.2** | **63.6** | **77.7** | 47.3 | 54.5 | **169.6** | 114.2 |
| Tired Actor (100 seq) | **87.5** | 50.1 | 33.1 | 61.4 | 81.7 | **46.9** | **61.4** | 172.7 | **111.9** |
| Baseline (1,000 seq) | 97.2 | **35.3** | **28.2** | 88.6 | **51.1** | 34.9 | 66.6 | 164.9 | 113.3 |
| Tired Actor (1,000 seq) | **98.4** | 40.6 | 29.5 | **90.0** | 55.3 | 36.1 | **67.9** | **163.4** | **109.1** |
| Baseline | 98.5 | **34.7** | 27.0 | 96.4 | 51.8 | 32.3 | 70.2 | 163.9 | 109.2 |
| Tired Actor | **99.1** | 37.4 | **25.4** | **97.9** | 47.6 | **30.1** | **71.4** | **163.3** | **103.4** |

Table 4: Quantitative results for fatigue-informed generalization.

**Left-Right Asymmetry.** As shown in Fig. 6, we initialize half of the human body with $M_F = 50\%$, and the other half with $M_F = 0\%$ when imitating a circle-running motion. The fatigue parameters are set as $F = 2, R = r = 0$. When the left (inner) half is fatigued, noticeable deviation is observed. With the fatigue accumulating, the character could even collapse. In contrast, the fatigue of the right (outer) half only introduces reduced action amplitude with marginal deviation. The asymmetric response to inner- and outer-half fatigue reflects the asymmetry of the underlying dynamic mechanism. Walking in circles might require the inner half of the body to exert higher torques to simultaneously maintain the gait and turn directions.

**Spatial Decision-Making.** Previous efforts (Brown et al., 2021; Daniels & Burn, 2023) demonstrated the correlation between energetic optima and spatial decision-making procedures like path selections. To identify whether similar behaviors exist for Tired Actor, we first distill Tired Actor into a student policy as $\pi_d(a^t|s^t_{task}, s^{self}_t)$ following DAGGER (Ross et al., 2011). Instead of the next-frame imitation target (Eq. 1) used as $s^{task}_t$ for the vanilla Tired Actor, $s^{task}_t$ is defined as: $s^t_{task} = \{kp_{task} - kp^t\}$, where $kp$ is the positions of a subset of human body keypoints, and $kp_{task}$ indicates a long-term target. In this way, the student policy is capable of selecting its own path towards a target pose, rather than strictly following the given path. To probe the spatial decision-making characteristics of Tired Actor, we put it in a simple path-following setting. The path is designed as a polyline with one turning angle and two target root locations.. Every three seconds, we switch to the next target root location. Each episode ends in six seconds. As in Fig. 7, Tired Actor avoids sharp turns and exert lower accelerations towards the changing target, resulting in lower trajectory curvatures. Also, we find Tired Actor managed to follow the trajectory more precisely compared to the baseline. However, When faced with rather sharp turns, Tired Actor could trade the target reaching timeliness off with smoother trajectory.

### 4.3 FATIGUE-INFORMED GENERALIZATION

The novel behaviors introduced by fatigue also extend the original distribution. To this end, we train and evaluate variants of the baseline and the Tired Actor by limiting the training data scale to 100 and 1,000 sequences. The evaluation is conducted on three datasets. The AMASS train set (11,313 sequences) is adopted for seen in-domain evaluation. The AMASS test set (140 sequences) is adopted for unseen in-domain evaluation. Besides AMASS, CIRCLE (Araujo et al., 2023) is adopted for unseen out-of-domain evaluation, given its 7,200 whole-body reaching motion sequences. As shown in Tab. 4, Tired Actor performs consistently better than the non-fatigue baseline in success rate on the seen AMASS train set, with slightly worse mPJPE-G. Also, the baseline is better at mPJPE-L

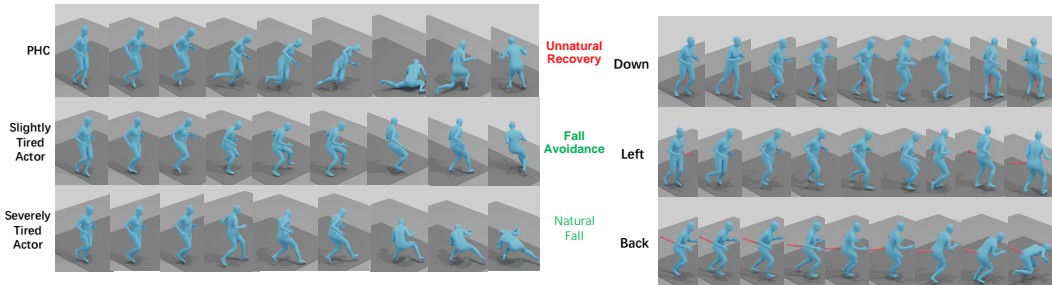

Figure 8: Fatigue-Induced Compensation under different $M_F$. Though PHC manages to recover after perturbation, unnatural gestures could be observed during recovery. Instead, Tired Actor adaptively responds to external perturbations with natural behaviors.

Figure 9: Fatigue-Induced Compensation under different directions. Lateral and downward perturbations are easier to absorb than backward perturbations.

with limited data scale, but the performance gap becomes smaller and smaller with the increasing training data scale. For the unseen in-domain evaluation on the AMASS test set, the baseline generally provides better performance with small-scale training data, but is finally surpassed by Tired Actor with sufficient data. When it comes to the unseen OOD evaluation on CIRCLE, both suffer from the domain gap. However, it is noticeable that Tired Actor is consistently better with different training data scales, demonstrating its better generalization ability.

From the quantitative results, we can observe a common macro trend: Tired Actor is equipped with a better generalization to handle unseen motion sequences, and the gain effect increases with the data scale. These trends coincide with our hypothesis that fatigue extends the data distribution. With more training data, better extensions could be made. We can also observe the trade-off of Tired Actor. Typically, Tired Actor prioritized the success rate, then mPJPE-L, and finally mPJPE-G. This manifests that by introducing fatigue, the controller is encouraged to keep balance and local pose resemblance, in exchange for less precise global pose tracking. Finally, an interesting observation is the surprising performance when trained on only 100 sequences. Even with very limited motion sequences, the controllers could provide competitive performance, indicating the inherent simplicity of human motor ability and inspire future works on data-efficient character control.

## 4.4 FATIGUE-INFORMED COMPENSATION

We further investigate the Tired Actor's ability to compensate for external perturbations. We apply a 400-Newton force for 1s at a certain part's centroid, starting from the same time step. More details are available in the appendix.

**Different Fatigue Levels.** In Fig. 8, we initialize actors with $MF = 0$ and $MF = 90$. The force is applied to the pelvis in the forward direction. As demonstrated, PHC falls and quickly recovers with unnatural motion. In contrast, the Tired Actor with initial $M_F = 0$ manages to resist the applied force and avoids falling. The Tired Actor with initial $M_F = 90$ falls with a more reasonable gesture. These suggest that by introducing fatigue, the Tired Actor is enhanced in its compensation ability.

**Different Force Directions.** In Fig. 9, the external force is applied at the pelvis from different directions of the global velocity direction. Backward perturbation forces are the most destabilizing, causing falls and deviations that challenge the recovery process. Lateral perturbations induce moderate instability, whereas downward forces are absorbed more effectively, resulting in minor disruption. These findings indicate anisotropic sensitivity to directional disturbances.

## 5 CONCLUSION

Inspired by insights from behavioral energetics, we introduced Tired Actor, a character controller incorporated with fatigue modeling, which maintained general motion coverage while achieving better naturalness. We further thoroughly investigated how the fatigue mechanism could influence the diversity, generalization, and robustness of character animation. We believe this work could open more possibilities in enhancing life-like character control.

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

APPENDIX

# A    EXTENDED EXEPRIMENTS

## A.1    IMPLEMENTATION DETAILS

The actor and critic networks are 7-layer MLPs with the hidden size of [2048, 1536, 1024, 1024, 512, 512] and SiLU activation. The AMP discriminator is a 3-layer MLP with a hidden size of [1024, 512] and ReLU activation. The policy is trained for 30K episodes, then finetuned with fall-recovery for 20K episodes. $\lambda_{ep}$ for the reweighted fatigue initialization is initially set as 1. It gradually increases to 3 in the first 20K episodes, then remains fixed at 3. IsaacGym (Makoviychuk et al., 2021) is adopted for physics simulation. The policy runs at 30Hz, while the simulation runs at 60Hz. All experiments are conducted on a single NVIDIA RTX3090 GPU with 1,536 concurrent environments. PPO is adopted for policy training, with a learning rate of 2e-5, discount factor $\gamma = 0.99$. For reward $r_{task}$, we set $\omega_1 = 0.5, \omega_2 = 0.1, \omega_3 = 0.3, \omega_4 = 0.1$. During training, we conduct an evaluation every 5k episodes to update the hard negative sampling weight.

## A.2    EXTENDED FATIGUE-INFORMED COMPENSATION

**Tired Actor with Different Fatigue Parameters** $F, R, r$**.** We find that a high $F$ causes the character to fall quickly, substantially degrading the performance. Quantitatively, keeping $R = 0.05, r = 1$ while increasing $F$ from 2 to 10 results in a 76.5% success rate on AMASS-Test. If we then increase the recovery rate $R$ from 0.05 to 0.2, the success rate could increase to 93.2%. However, the rest recovery multiplier $r$ makes less contribution compared to $F$ and $R$, and barely influences the success rate. Qualitative demonstrations are included in the supplementary video.

**Tired Action with Different perturbation parts.** In Fig. 10, perturbations are applied to different body parts. Perturbations applied to distal segments like hands are easily handled with little trajectory deviation, suggesting that disturbances in peripheral regions are more readily compensated for. In contrast, forces on central regions like the torso induce instability and falls. This pattern reflects the importance of maintaining core stability within the control policy.

## A.3    ABLATION STUDIES

**Hard-Negative Mining.** Tired Actor without hard-negative mining achieves 95.2% success rate, 43.2 mPJPE-G, and 30.5 mPJPE-L on AMASS-Train. On AMASS-Test, it achieves a 92.3% success rate, 52.5 mPJPE-G, and 35.1 mPJPE-L. The performance degradation indicates the efficacy of focusing more on learning from hard sequences.

**Fall-Recovery Finetuning.** Fig. 11 illustrates the Tired Actors with and without fall-recovery fine-tuning imitating a forward-walking motion with initial $M_F = 95\%$. Tired Actor managed to avoid falling, though the gesture is twisted to some extent. Without fall-recovery fine-tuning, the character fails to follow the reference motion and falls on the ground. This reflects that the fall-recovery fine-tuning helps improve the robustness of the controller.

# B    DISCUSSION

Although Tired Actor achieves better naturalness while maintaining the wide motion coverage, some limitations remain. Compared to (Cheema et al., 2023), the emerging behavior patterns of Tired Actor are mostly limited to reduced action amplitude, while other expected motor strategies, like waiting, are rarely observed. We identify this as a result of our strict task setting of per-frame motion imitation. With the reference motion as a strong constraint, the Tired Actor struggles and falls rather than waiting for recovery from fatigue. Therefore, extending Tired Actor to less-strict scenarios would be a promising goal. Additionally, the generalization ability of Tired Actor, although impressive, remains limited for out-of-domain data. Given the fatigue-informed data distribution extension, it would be interesting to exploit the generalization potential of motion control further. Finally, though the Tired Actor is enhanced in natural compensation against external perturbations, unnatural gestures could still be observed. Also, the behavior after falling could collapse. There are still substantial spaces for reducing the unnaturalness in character control.

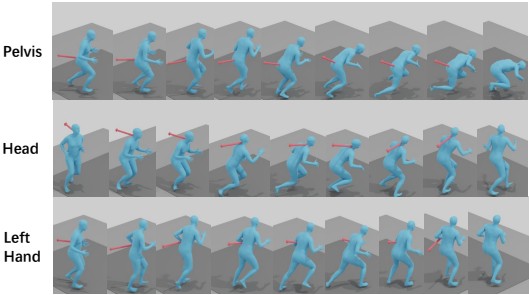

Figure 10: External perturbation compensation with different perturbation parts. Perturbations applied to distal segments are easier to handle than those applied to central body parts.

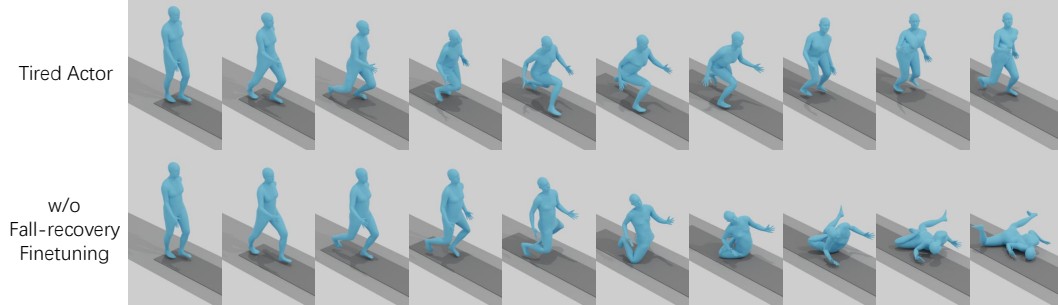

Figure 11: Tired Actors with and without fall-recovery fine-tuning imitate a forward-walking motion with initial $MF = 95\%$. The fall-recovery fine-tuning enhances the robustness of the Tired Actor and prevents it from falling.