# OpenReview forum: "Tired Actor: Fatigue-Informed Character Control"
_ICLR.cc/2026/Conference — Submitted to ICLR 2026_

### Official Review · Reviewer_ZZH3 · 2025-10-25

**Soundness:** 2
**Presentation:** 3
**Contribution:** 2
**Rating:** 2
**Confidence:** 4

**Summary:**

The paper proposes Tired Actor, a fatigue-informed and enery-aware character controller that incorporates human-like fatigue machenism into physics-based motion imitation. By adapting the Three-Compartment Controller model, it modulates maximal torques to simulate fatigue, encouraging more natural and diverse behaviors. Experiments show that fatigue can improve local tracking performance and generalization to unseen motions, and compensation strategies in controlled characters.

**Strengths:**

1. The paper presents an interesting and novel idea by introducing fatigue as a proxy for energy-awareness in character control, bridging insights from behavioral energetics with physics-based animation.

2. The authors provide extensive behavioral analysis, thoroughly demonstrating how fatigue influences the character behaviors, generalization capabilities and other compensations. The experiments are comprehensive and offer clear comparisons and convincing illustrations of the effects.

**Weaknesses:**

1. The practical value of Tired Actor is unclear. While it introduces some interesting, human-like fatigue behaviors, it remains questionable whether such a precise fatigue mechanism is actually necessary. 1) From the animation perspective, Tired Actor sacrifices global tracking performance for subtle local fatigue effects. Even in out-of-domain cases, it performs worse than PHC. The qualitative results in Fig. 1 (e.g., the jump and nearly immobile left leg) do not convincingly suggest more natural motion. 2) From energy efficiency perspective: In robotics and control, energy efficiency is usually improved through simple regularizations such as penalizing torque or action rate (He et al., 2025a) a. The proposed fatigue modeling does not demonstrate a better trade-off between tracking accuracy and energy consumption or the adaptive behavior among different motion tracking. Since human muscle fatigue mechanisms differ greatly from robots, it is unclear why a human-like fatigue model should be applied here without clear evidence of efficiency or performance benefits.

2. The experimental validation of the proposed methods is still insufficient. First, the paper lacks comparisons with simple baselines, such as directly limiting joint outputs to 20% of maximum or adding a soft penalty, so the claimed benefits of Tired Actor are not clearly demonstrated. Additionally, a clear curve showing the evolution of MF during motion play would better illustrate how Tired Actor actually works. Second, the effectiveness of the reweighted fatigue initialization is not convincingly shown, as there is no direct comparison with uniform initialization to support its contribution. Third, the authors claim that the resulting motion is more natural, but there is no objective comparison such as a user study, it is actually difficult to convince others.

3. Others related to analysis in experiments part: 1) The dependence on data size raises questions. If the fatigue mechanism indeed introduces distributional variety, it is unclear why Tired Actor generalizes poorly on small-scale data while performing better in-domain. 2) Additionally, the part-wise analysis, although showing some human-like behaviors, does not seem inherently related to the fatigue mechanism—for example, left-right asymmetry could likely emerge from a simple asymmetric torque penalty.

**Questions:**

1. I am quite curious about the failure cases of PHC and the baseline. Specifically, in which types of motions does the Tired Actor achieve a higher success rate? Understanding this could help clarify why Tired Actor exhibits better generalization.

2. I am not sure if I understand correctly, but if the Tired Actor performs poorly in global tracking, why is its pelvis tracking performance better than the baseline?

3. The performance still seems sensitive to MF initialization. I wonder whether different motions exhibit different sensitivities—for example, jumping may rely more on leg control, while dancing may require more active arm movements. If a single MF initialization cannot work well across all motions, how should the method handle training on a large dataset containing diverse and challenging motions?

---

### Official Review · Reviewer_RUyh · 2025-10-31

**Soundness:** 3
**Presentation:** 3
**Contribution:** 3
**Rating:** 6
**Confidence:** 3

**Summary:**

This paper addresses the unnatural motion issue in physics-based character control caused by the lack of biomechanical and physiological priors. Inspired by behavioral energetics, it proposes Tired Actor, a fatigue-informed character controller that incorporates the Three-Compartment Controller (3CC) model to simulate fatigue via modifying the maximal applicable torque. The method maintains general motion coverage while enhancing naturalness, and systematically investigates how fatigue influences motion diversity, generalization, and perturbation compensation. Key technical contributions include reweighted fatigue initialization (addressing imbalanced fatigue distribution), hard negative mining (focusing on difficult sequences), and fall-recovery finetuning (improving robustness). Experiments on the AMASS and CIRCLE datasets show that Tired Actor outperforms baselines in success rate, generalization to unseen motions, and natural compensation. However, it trades off global pose-tracking precision for local naturalness.

**Strengths:**

1. Pioneers the incorporation of fatigue (as an energy-awareness proxy) into general character control, bridging behavioral energetics with physics-based animation—filling a gap in existing data-driven methods that ignore energy constraints.

2. The 3CC-based fatigue modeling, combined with reweighted initialization, hard negative mining, and fall-recovery finetuning, forms a rigorous pipeline that addresses practical challenges (e.g., imbalanced fatigue distribution, fragile recovery).

3.  Conducts in-depth studies on fatigue’s impact across motion imitation, behavioral diversity (e.g., asymmetric responses, spatial decision-making), generalization, and perturbation compensation—providing actionable insights for future work.

4. Demonstrates tangible gains in motion naturalness (e.g., replacing unnatural foot sliding with leg swinging, adaptive action amplitude reduction) that align with human movement patterns

**Weaknesses:**

1. Tired Actor underperforms baselines in global pose error (mPJPE-G), velocity, and acceleration imitation, indicating a suboptimal balance between local naturalness and global trajectory fidelity.
2. The novel behaviors introduced by fatigue are mostly restricted to reduced action amplitude; more human-like strategies (e.g., pausing to recover) are not observed due to strict per-frame motion imitation constraints.
3. While outperforming baselines on CIRCLE, Tired Actor still suffers from domain gaps, suggesting the fatigue-extended data distribution has limited ability to cover highly diverse out-of-domain motions.

**Questions:**

1. What is the empirical basis for choosing fatigue parameters (F=2, R=0.05, r=1)? How do these parameters generalize to different motion types (e.g., high-intensity sports vs. low-intensity walking)?
2.  How does Tired Actor compare to muscle-actuated fatigue controllers (e.g., Feng et al., 2023) in terms of naturalness, computational efficiency, and scalability to complex motions?
3. Can the model provide interpretable links between specific fatigue states (e.g., muscle group-specific fatigue) and observed motion adjustments? This would enhance trustworthiness for practical applications (e.g., VR, robotics).

---

### Official Review · Reviewer_ZnNG · 2025-11-01

**Soundness:** 2
**Presentation:** 3
**Contribution:** 1
**Rating:** 2
**Confidence:** 3

**Summary:**

In this work, the author proposed to Leveraging the Three-Compartment Controller (3CC) model,
to guardrail against the actor generate unrealistic and unphysically plausible motions.
This is a common artifact when the reward modeling is heavily biased towards the minimizing the distance between the actor and the target pose.

In summary, the 3CC model enables the learning of a fatigue-aware policy by dynamically constraining the actor's joint torque limits.
Limits are enforced in a nonlinear, state- and time-dependent manner.

**Strengths:**

1. The paper addresses an interesting problem of learning natural and robust character control with fatigue.

2. The paper has a comprehensive experimental analysis.

A complete ablation study on the fatigue levels,
applications to different body parts and spatial decision-making are conducted.
And the algorithm is tested in different dataset and tested against unseen data and perturbation.

3. The paper is well written and the figures are clear and informative.

4. The authors are transparent about the limitations of their approach.

**Weaknesses:**

1. Limited behavioral diversity is shown with the experiments section and in the demo.
Most experiments are conducted for natural recovery around the ground motions (getting up from the ground, avoiding falling to the ground),
Or force perturbations.

It is unclear if the proposed method performs well for more complex behaviors like landing on the ground, colliding into a wall etc.
This limitation makes the proposed method unclear for general application.

2. The assumption that we need fatigue is not necessarily true, as the unnatural recovering artifact has been addressed by other methods.

For example AMP [1, 2] generally does not suffer from the unnatural recovering artifact.
In ASE [3] the recovering artifast exists but this is because there were no getting up motions in the first place.

I believe the studied artifact is strongly associated to motion tracking formulation where reward modeling which
model the reward as the exponential negative distance between the actor and the target pose.

But for tracking policy,
it will make more sense to have a separate planning module which generates the natural recovering motions in the first place.

Even in real robots getting up naturally has been solved in a similar way as mentioned above.
For example https://bfm-zero-anonymous.pages.dev/#/

[1] Hassan, Mohamed, Yunrong Guo, Tingwu Wang, Michael Black, Sanja Fidler, and Xue Bin Peng.
"Synthesizing physical character-scene interactions." In ACM SIGGRAPH 2023 Conference Proceedings, pp. 1-9. 2023.

[2] Peng, Xue Bin, Ze Ma, Pieter Abbeel, Sergey Levine, and Angjoo Kanazawa.
"Amp: Adversarial motion priors for stylized physics-based character control." ACM Transactions on Graphics (ToG) 40, no. 4 (2021): 1-20.

[3] Peng, Xue Bin, Yunrong Guo, Lina Halper, Sergey Levine, and Sanja Fidler.
"Ase: Large-scale reusable adversarial skill embeddings for physically simulated characters." ACM Transactions On Graphics (TOG) 41, no. 4 (2022): 1-17.

3. The novelty from previous work is rather incremental and does not seem to meet the threshold for ICLR.

In [4], a largely similar idea has been proposed but used for different application.

[4] Cheema, Noshaba, Rui Xu, Nam Hee Kim, Perttu Hämäläinen, Vladislav Golyanik, Marc Habermann, Christian Theobalt, and Philipp Slusallek.
"Discovering fatigued movements for virtual character animation." In SIGGRAPH Asia 2023 Conference Papers, pp. 1-12. 2023.

**Questions:**

1. The demo video's voice is cut off at the beginning.

---

### Author Response · Authors · 2025-12-04

Dear Area Chairs and Reviewers:

Thanks for your valuable reviews and insightful comments, which would help us improve our paper.

In general response, we would like to clarify our stance on some major concerns.

- Limited diversity (by ZnNG W1 and RUyh W2). We agree that in its current form, the Tired Actor's behavior could be limited to the over-strict motion-tracking setting. Extending it with more flexible paradigms, such as AMP and BFM, would be promising for future revisions.

- Limited practical value (by ZnNG W2, RUyh W1 and ZZH3 W1-2). We would like to clarify that introducing bio-inspired mechanisms like fatigue is aimed at exploring the potential of human-likeness instead of achieving stronger capability. We agree that, given its current form and the long-term absence of feasible naturalness metrics beyond the user study, further refinement is required to validate this method's effectiveness.

The rest suggestions, especially those related to experimental validations, would be carefully considered and incorporated in future revisions.

We thank the reviewers and ACs again for their time and efforts!

---

### Meta-Review · Area_Chair_bdFr · 2026-01-04

**Summary:**

The primary concerns centered on the limited behavioral diversity shown, the unclear practical value or necessity of the fatigue mechanism (especially due to the trade-off with global tracking accuracy), and the insufficient experimental validation (missing simple baselines and objective naturalness metrics like a user study).

The authors acknowledge these limitations as requiring future work but did not resolve them in the rebuttal. Therefore, the paper is rejected.

**Reviewer Concerns:**

The authors acknowledged these concerns raised by reviewers but did not resolve them in the rebuttal.

**Reviewer Scores:**

The authors acknowledged these concerns raised by reviewers but did not resolve them in the rebuttal.

---

### Decision · Program_Chairs · 2026-01-26

Reject